# Identification of New Antifungal Agents Targeting Chitin Synthesis by a Chemical-Genetic Method

**DOI:** 10.3390/molecules24173155

**Published:** 2019-08-29

**Authors:** Yan Li, Hongmin Sun, Xiaohong Zhu, Cong Bian, Yanchang Wang, Shuyi Si

**Affiliations:** 1Beijing Key Laboratory of Antimicrobial Agents, Institute of Medicinal Biotechnology, Peking Union Medical College and Chinese Academy of Medical Sciences, Beijing 100050, China; 2Department of Biomedical Sciences, College of Medicine, Florida State University, Tallahassee, FL 32306, USA

**Keywords:** chitin, glucan, *Saccharomyces cerevisiae*, *Candida albicans*, antifungal agents

## Abstract

Fungal infection is a leading cause of mortality in immunocompromised population; thus, it is urgent to develop new and safe antifungal agents. Different from human cells, fungi have a cell wall, which is composed mainly of polysaccharide glucan and chitin. The unique cell wall structure is an ideal target for antifungal drugs. In this research, a chemical-genetic method was used to isolate antifungal agents that target chitin synthesis in yeast cells. From a compound library, we isolated two benzothiazole compounds that showed greater toxicity to yeast mutants lacking glucan synthase Fks1 compared to wild-type yeast cells and mutants lacking chitin synthase Chs3. Both of them inhibited the activity of chitin synthase in vitro and reduced chitin level in yeast cells. Besides, these compounds showed clear synergistic antifungal effect with a glucan synthase inhibitors caspofungin. Furthermore, these compounds inhibited the growth of *Saccharomyces cerevisiae* and opportunistic pathogen *Candida albicans*. Surprisingly, the genome-wide mass-spectrometry analysis showed decreased protein level of chitin synthases in cells treated with one of these drugs, and this decrease was not a result of downregulation of gene transcription. Therefore, we successfully identified two new antifungal agents that inhibit chitin synthesis using a chemical-genetic method.

## 1. Introduction

Fungal infection is a leading cause of mortality in the immunocompromised population, such as patients who undergo anticancer chemotherapy, radiation therapy, parenteral nutrition, or organ transplant [1]. This population also includes patients infected with HIV, which disrupts the immune system [2]. In contrast to the increased incidence of fungal infections, the number of efficient and less toxic antifungal antibiotics is limited. Also, the increased use of antifungal agents leads to the emergence of drug-resistant fungal strains [3,4]. Therefore, it is urgent to develop new and safe antifungal agents.

Unlike human cells, fungi have a cell wall structure, which is an ideal target for antifungal drugs. The fungal cell wall is highly dynamic and essential for cell viability, morphogenesis, and pathogenesis. Although the components of the fungal cell wall vary among species, most fungal cell wall consists of a core of β-1,3-glucan polysaccharide covalently cross-linked to chitin, forming the primary scaffold structure. As a key component of the fungal cell wall, chitin contributes to about 2% (*w*/*w*) of fungal dry weight [5]. Chitin is a polysaccharide composed of β-(1, 4) N-acetyl-D-glucosamine (GlcNAc). During cell wall synthesis, chitin connects to β-1,3-glucan and β-1,6-glucan to form higher-order complexes, which in turn link to mannan and cell wall proteins [6,7,8]. During cell division, chitin plays a vital role in chitin ring synthesis at the bud neck and in the formation of primary septum [9,10].

Budding yeast *S. cerevisiae* contains three chitin synthases (CS), CSI, CSII, and CSIII, and the genes encoding these three enzymes are *CHS1*, *CHS2*, and *CHS3,* respectively. All these three genes are nonessential, but deletion of both *CHS2* and *CHS3* in budding yeast leads to cell death [11,12]. Chs1 is believed to be responsible for repairing the chitin septum during cytokinesis [9]. Chs2 is necessary for chitin synthesis at the primary septum, and deletion of *CHS2* gene results in abnormal bud morphology [10]. Chs3 enzyme contributes to the synthesis of most chitin in the cell wall during bud emergence and growth, mating, and spore formation [13]. Chitin synthase enzymes are synthesized in the cytoplasm and then transported to the cell membrane for chitin synthesis. The localization of Chs3 changes during the cell cycle, which is regulated by additional chitin synthesis-related proteins, Chs4-7. Chs3 forms a complex with Chs4/Skt5, and Bni4 protein localizes this complex to the septin ring at the bud neck. Chs7 is required for the dissociation of Chs3 from the endoplasmic reticulum, while Chs5 and Chs6 are involved in the transport of Chs3 from the trans-Golgi network to plasma membrane [14].

For most fungal species, β-1,3-glucan is the main polymer of the cell wall, comprising between 65 and 90% of the whole cell wall [15,16]. β-1,3-glucan is synthesized by a membrane-associated glucan synthase complex, which uses UDP-glucose as a substrate. Fks1 and Fks2/Gsc2 are large integral membrane proteins that catalyze β-1,3-glucan synthesis. Rho1 is a small GTPase protein, which enhances the enzyme activity of Fks1 and Fks2 [17]. Yeast cells lacking *FKS1* gene are still viable, but the combination of *fks1*∆ with mutants lacking some chitin synthesis-related genes, such as *chs3*∆, *chs5*∆, *chs6*∆, and *chs7*∆, causes synthetic lethality [18]. These genetic interactions support the idea that simultaneous reduction in β-1,3-glucan and chitin synthesis in yeast cells cause viability loss.

Some antifungal agents target cell wall structure. Nikkomycin and polyoxin are specific chitin synthase inhibitors, and nikkomycin Z is under clinical trial for the treatment of fungal infections, such as coccidioidomycosis [19]. Echinocandins inhibit β-1,3-glucan synthesis in the cell wall, and this group of agents is wildly used for the treatment of fungal infections caused by yeast, such as *Candida* species, or molds, such as *Aspergillus* [20]. A recent study indicated a synergistic effect for the combination of echinocandins and nikkomycin Z against infections caused by *C. albicans* using a mouse model [21]. Therefore, chitin inhibitors could be used in combination with enchinocandins for the treatment of fungal infections.

To screen antifungal cell wall agents, previous studies used purified chitin and glucan synthases to isolate compounds that inhibit their enzyme activity in vitro [22,23], but the results might not reflect the antifungal activity in vivo. In this study, we used a chemical-genetic method to isolate antifungal agents that impair chitin synthesis in yeast cells. This idea stems from the synthetic lethality between yeast mutants lacking the glucan synthase gene (*fks1∆*) and mutants with compromised chitin synthesis [18]. We speculated that *fks1*∆ mutant is more sensitive to chitin synthesis inhibitors than wild-type (WT) cells or chitin synthase mutants, such as *chs3∆*. Based on this idea, we isolated two compounds that showed increased toxicity to *fks1*∆ yeast mutants compared to WT and *chs3∆* mutant cells. These two compounds inhibited chitin synthesis and reduce chitin level in yeast cells. Using whole-cell extract, we found that they inhibited the activity of chitin synthase. Also, the genome-wide mass-spectrometry analysis showed decreased protein level of chitin synthases in cells treated with one of these drugs, but this decrease was not caused by the alternation of gene transcription. The compounds also exhibited growth inhibition of budding yeast and human pathogen *C. albicans* and showed the clear synergistic effect with glucan synthase inhibitors caspofungin, an echinocandin derivative. Therefore, we successfully identified new antifungal agents using a chemical-genetic approach.

## 2. Results

### 2.1. To Screen Agents that Are More Toxic to Yeast Glucan Synthase Mutants

Glucan and chitin are the two major components of the fungal cell wall. In budding yeast *S. cerevisiae*, Fks1 protein is the catalytic subunit of β-1,3-glucan synthase, while chitin synthase Chs3 catalyzes chitin synthesis. Although deletion of either *FKS1* or *CHS3* in budding yeast does not lead to cell death, yeast cells lacking both genes cannot survive [18], which supports the idea that simultaneous reduction in β-1,3-glucan and chitin synthesis cause cell death. If that is the case, yeast mutants with impaired chitin synthesis should be more sensitive to the antifungal drugs targeting β-1,3-glucan synthesis than WT cells and mutants with compromised glucan synthesis. Similarly, yeast mutants with impaired glucan synthesis should be more sensitive to the antifungal drugs targeting chitin synthesis than WT cells and mutants with compromised chitin synthesis.

As a proof of concept, we first assessed the growth inhibition of chitin synthase inhibitor nikkomycin Z and glucan synthase inhibitor caspofungin to WT *S. cerevisiae* strain BY4741 and mutants with the compromised synthesis of glucan (*fks1∆*) or chitin (*chs3∆*). As a result, nikkomycin Z did not show growth inhibition to WT and *chs3∆* mutant strains until the concentration reached to 200 μg/mL, but for *fks1∆* mutant strain, the minimum inhibitory concentration (MIC) of nikkomycin Z was 25 μg/mL (Table 1). Similarly, the MICs of glucan synthase inhibitor caspofungin for WT and *fks1∆* were 0.03 μg/mL and 0.015 μg/mL, respectively, but the MIC was 0.00375 μg/mL for *chs3∆*. Therefore, yeast cells with compromised glucan synthesis were more sensitive to chitin synthase inhibitors nikkomycin Z. In contrast, yeast cells with impaired chitin synthesis were more sensitive to glucan synthesis inhibitor caspofungin. Besides, nikkomycin Z and caspofungin showed synergistic antifungal effect for WT yeast cells (Table 2). Together, these results indicate the feasibility to screen antifungal agents that target cell wall using yeast mutants with impaired synthesis of chitin (*chs3∆*) or glucan (*fks1∆*).

For this screening, the growth of WT, *chs3∆,* and *fks1∆* yeast cells was assessed in 96-well plates in the presence of compounds at 100 μg/mL. The compounds that inhibited the growth of *fks1∆* mutant cells at lower concentration compared to WT and *chs3∆* were selected as potential chitin synthesis inhibitors. In contrast, the compounds that showed more significant growth inhibition to *chs3∆* than WT and *fks1∆* were likely glucan synthesis inhibitors. The compounds that inhibited the growth of all these strains at 100 μg/mL were further assessed at lower concentrations to clarify their differential inhibitory activity toward these strains. With this method, we screened 50,000 compounds from a compound library, a combination of synthetic compounds purchased from Enamine (Kyiv, Ukraine) and Life Chemicals Inc. (Burlington, ON, Canada) and natural products from the Institute of Medicinal Biotechnology. Among them, five compounds showed greater growth inhibition of *fks1∆* mutants including IMB-D10 and IMB-F4 (Table 1), and four exhibited greater growth inhibition of *chs3∆.* This work was focused on the antifungal activity of the first group of compounds, which might inhibit chitin synthesis.

### 2.2. The Effect of the Candidate Compounds on Yeast Chitin Levels

In budding yeast *S. cerevisiae*, chitin distributes all over the cell wall and is enriched at the bud-neck [13]. To further test whether the five candidate compounds that show greater growth inhibition on *fks1∆* mutants impair chitin synthesis, the chitin level in WT yeast cells treated with these compounds was assessed after staining with Calcofluor White M2R (CFW), a fluorescent dye that specifically binds to chitin. Yeast cells were treated with 12.5 or 25 μg/mL compounds for 3 or 5 hours and stained with CFW. Compared with control cells treated with DMSO, cells treated with IMB-D10 and IMB-F4 showed weaker fluorescent signal around the cell wall and at the primary septum (Figure 1A). We further analyzed the effect of compounds IMB-D10 and IMB-F4 on chitin content using Morgan–Elson method, as described [22]. The chitin level in yeast cells treated with these two compounds reduced significantly compared to control cells (Figure 1B). As a test of the methodology, we found that the chitin content of *chs3∆* mutant cells was 56.74 ± 8.24% of that in WT cells, indicating the feasibility of this method. In yeast cells treated with 25 μg/mL IMB-D10, the chitin content decreased to 47.61 ± 6.79% compared to cells treated with DMSO, and the difference was significant. However, the treatment of yeast cells with 12.5 μg/mL IMB-D10 did not reduce the chitin level significantly. For cells treated with 12.5 or 25 μg/mL IMB-F4, the chitin content reduced to 46.09 ± 8.01% and 27 ± 5.33%, respectively (Figure 1B). The effect of the other three compounds on chitin level in yeast cells was also examined with CFW staining and Morgan–Elson method, but no significant reduction of chitin level was detected. Therefore, compounds IMB-D10 and IMB-F4 were selected for further study, and their structures are shown in Figure 1C. The structure of IMB-D10 is *N*-(2-(diethylamino) ethyl)-*N*-(7-hydroxy-4-methoxybenzo[d]thiazol -2-yl)-2-oxo-2*H*-chromene-3-carboxamide, and the structure of IMB-F4 is *N*-(2-(1*H*-imidazol-1-yl)ethyl)-3,5-dimethoxy-*N*- (6-methoxybenzo[d]thiazol-2-yl)benzamide. Both compounds were from the compound library synthesized by Life Chemicals Inc. (Burlington, ON, Canada).

### 2.3. IMB-D10 and IMB-F4 Inhibit Fungal Growth at High Concentrations without Toxicity to Human Cells

The MIC of compound IMB-D10 for budding yeast *S. cerevisiae* was 50 μg/mL. IMB-F4 did not inhibit cell growth of *S. cerevisiae* at 100 μg/mL. However, yeast cells exhibited dose-dependent slow growth in the presence of these two compounds by measuring the OD (optical density) of yeast cells over time (Figure 2). We also determined the toxicity of these two compounds to human embryonic kidney 293 cells and Hela cells. No growth inhibition was noticed when the concentration of these compounds was as high as 100 μg/mL. Besides, these two compounds did not inhibit the growth of bacteria *Escherichia coli* ATCC 25922 and *Mycobacterium smegmatis* at 100 μg/mL, indicating that these compounds specifically inhibit the growth of yeast cells.

### 2.4. The Effect of IMB-D10 and IMB-F4 on the Activity of Antifungal Agents Targeting Cell Wall

CFW exhibits antifungal activity by binding to nascent chitin chains and preventing the connection between chitin and glucan [24]. Previous results show that *chs6∆* yeast mutant is more resistant to CFW, indicating that decreased chitin synthesis likely reduces the antifungal activity of CFW [25]. The MIC of CFW against budding yeast was 6.25 μg/mL, but the MICs of CFW increased in the presence of IMB-D10 in a dose-dependent manner from 6.25 μg/mL to 100 μg/mL. For example, the presence of 12.5 μg/mL IMB-D10 caused an eight-fold increase of the MIC of CFW against budding yeast (Figure 3A). Yeast cells treated with 25 and 50 μg/mL IMB-F4 also increased resistance to CFW, but the effect was less significant compared to IMB-D10.

Glucan synthase inhibitor caspofungin and chitin synthase inhibitor nikkomycin Z showed synergistic antifungal activity against budding yeast *S. cerevisiae* (Table 2). We speculated that other chitin synthase inhibitors would also show synergistic antifungal activity with caspofungin. Indeed, IMB-D10 and IMB-F4 lowered the MIC of caspofungin against budding yeast. The fractional inhibitory concentration index (FICI) for IMB-D10 and IMB-F4 were equal to or less than 0.5, which is considered to have a synergistic effect (Table 2).

Yeast cells lacking Chs1, Chs2, and Chs3 form chain-like structure due to cytokinesis failure [26]. If IMB-D10 inhibits chitin synthesis, we expected that yeast cells treated with this compound would show abnormal morphology. After treatment with 25 μg/mL IMB-D10 for 5 hours at 30 °C, some yeast cells failed to divide, resulting in cell clusters with more than two cell bodies (Figure 3B). However, it appeared that yeast cells treated with IMB-F4 showed normal cell separation. We speculated that IMB-D10 would likely show stronger inhibition of chitin synthesis, which would lead to defective cell separation. Together, these results suggested that these two compounds impaired chitin synthesis in yeast cells.

### 2.5. Inhibition of Chitin Synthase Activity by IMB-D10 and IMB-F4

Our preceding results indicated that compounds IMB-D10 and IMB-F4 reduced chitin level in yeast cells. One possibility is that these two compounds inhibit the enzyme activity of chitin synthases. To test this idea, we analyzed the effect of these two compounds on the activity of chitin synthases. We prepared Chs1, Chs2, and Chs3 from yeast extracts, as described previously [22], and the chitin synthesis activity of these enzymes was examined in the presence and absence of these compounds. IMB-D10 inhibited chitin synthesis with half-inhibitory concentration(IC_50_) values of 17.46 ± 3.39, 3.51 ± 1.35, and 13.08 ± 2.08 μg/mL for Chs1, Chs2, and Chs3, respectively (Figure 4A–C). However, IMB-F4 only inhibited the activity of Chs2 and Chs3 with the IC_50_ of 8.546 ± 1.42 μg/mL and 2.963 ± 1.42 μg/mL (Figure 4D,E). Therefore, both IMB-D10 and IMB-F4 likely act as chitin synthase inhibitors.

### 2.6. IMB-D10 Reduces the Protein Level of Three Chitin Synthases

To further study the mechanism by which IMB-D10 and IMB-F4 inhibit the growth of yeast cells, we used isobaric tags for relative and absolute quantification (iTRAQ) technology to compare the level of each yeast protein before and after treatment with these two compounds. Surprisingly, we detected a significant decrease of the three chitin synthases after treatment with IMB-D10. Compared with cells treated with DMSO, the relative abundance of Chs1, Chs2, and Chs3 in cells treated with 25 μg/mL IMB-D10 was 22 ± 5.10%, 21.33 ± 6.18%, and 18.33 ± 1.89%, respectively. In clear contrast, the relative abundance of these synthases in cells treated with 25 μg/mL IMB-F4 remained similar compared with DMSO treated cells (Figure 5A). To clarify if the decrease of protein levels is a result of transcriptional downregulation, we examined the expression levels of *CHS1*, *CHS2,* and *CHS3* genes using the qRT-PCR method. The expression of these three genes did not decrease significantly as their cognate proteins (Figure 5B), indicating that the decreased protein levels of these three enzymes were unlikely a result of compromised gene expression. The reason that IMB-D10 downregulates the protein levels of the three chitin synthases in yeast cells is currently unknown. One possible explanation is that IMB-D10 binds to chitin synthases, which not only inhibits their enzyme activity but also destabilizes these proteins.

### 2.7. IMB-D10 and IMB-F4 Inhibit the Growth of Candida albicans

We further examined the effect of these two compounds on the growth of opportunistic human pathogen *C. albicans.* The MIC of compound IMB-D10 for *C. albicans* ATCC10231 was 100 μg/mL. Like budding yeast, the MIC of IMB-F4 was more than 100 μg/mL. However, the growth of *C. albicans* was reduced after 4 hours incubation in the presence of 50 μg/mL and 100 μg/mL of IMB-D10. The growth inhibition is dose-dependent and significant (*p* < 0.05). For *C. albicans* incubated with IMB-F4 at 50 μg/mL and 100 μg/mL, growth inhibition was observed after incubation for 6 hours (*p* < 0.05). These results revealed that both IMB-D10 and IMB-F4 inhibited the growth of *C. albicans*, but IMB-D10 exhibited stronger inhibitory effect.

We also analyzed if the MIC of CFW against *C. albicans* changes in the presence of various concentrations of nikkomycin Z and these two compounds. In the absence of chitin synthesis inhibitor, the MIC of CFW against *C. albicans* was 12.5 μg/mL, but the MIC of CFW increased to 100 μg/mL in the presence of 3.125 μg/mL nikkomycin Z. Similarly, a dose-dependent increase of MIC of CFW was observed in the presence of IMB-D10. For example, the MIC of CFW increased to 100 μg/mL by the presence of IMB-D10 at 50 μg/mL. The presence of 25 and 50 μg/mL IMB-F4 also increased the resistance of *C. albicans* to CFW, but the effect was less significant compared to IMB-D10 (Figure 6B). Besides, the results of CFW staining showed decreased chitin level in *C. albicans* treated with these two compounds (Figure 6C). On the other hand, the synergistic antifungal activity against *C. albicans* was detected between caspofungin and these two compounds. The presence of IMB-D10 and IMB-F4 at 12.5 μg/mL lowered the MIC of caspofungin against *C. albicans* from 0.025 μg/mL to 0.00625 and 0.00315 μg/mL, respectively (Table 3). Together, these results suggest that these two compounds likely inhibit chitin synthesis in fungal cells.

## 3. Discussion

Here we used a chemical-genetic method to screen agents that inhibit chitin or glucan synthesis in yeast cells. This screening is based on the observation that the combination of yeast mutants with impaired chitin synthesis, such as *fks1∆*, with mutants showing impaired glucan synthesis (*chs3∆*), leads to cell death. We speculated that *fks1∆* mutant cells would be more sensitive to chitin synthesis inhibitors. With this idea, we isolated two compounds that inhibited chitin synthesis. We also identified some compounds that might inhibit glucan synthesis (data not shown), but these compounds would be of our research interest in the future. These results have demonstrated the feasibility of this screening method for antifungal compounds that inhibit the synthesis of cell wall components—chitin or glucan.

One problem for the screening of chitin synthesis inhibitors is that these inhibitors, such as nikkomycin, exhibit very high MIC against budding yeast *S. cerevisiae*. Moreover, in *fks1∆* mutant cells, chitin content increases as a result of compensation [27,28], which likely compromises the sensitivity to chitin synthesis inhibitors. Therefore, we used a high compound concentration (100 μg/mL) during the initial screening. Moreover, compounds with two-fold or higher MIC difference between *fks1∆* and *chs3∆* or WT cells were selected so that few positive compounds were missed. Because glucan is the major cell wall component in yeast cells, the defect of the cell wall structure in *fks1∆* is likely more serious than that in *chs3∆* cells, which may make *fks1∆* cells more sensitive to some compounds due to the higher permeability. Also, *fks1∆ fks2∆* double mutants in *S. cerevisiae* and *Candida glabrata* are lethal [29,30]; thus, we cannot exclude the possibility that a compound that is more toxic to *fks1∆* could be an inhibitor of Fks2 enzyme based on the result of initial screening. Therefore, we performed more experiments to distinguish these possibilities.

The inhibition of chitin synthesis and antifungal activity of IMB-D10 and IMB-F4 have never been reported. These two compounds are benzothiazole with some similarities in structure, but their activity in the inhibition of chitin synthesis is likely different. IMB-D10, but not IMB-F4, inhibited Chs1 activity and reduced the protein level of the three chitin synthases. Besides, IMB-D10 showed higher inhibitory activity against *C. albicans* than IMB-F4. These results indicate that their structural difference might have a greater impact on their activity. Compared with the structure of IMB-F4, a chlorine substituent on the benzene ring of benzothiazole and a basic aliphatic tertiary amine in the IMB-D10 likely contribute to the different antifungal activity of these two compounds. In-depth understanding of the structure-activity relationship of this class of compounds needs systematic analysis of the antifungal activity and mechanism of a series of similar compounds.

An unexpected result is that IMB-D10 significantly reduced the protein level of three chitin synthases in yeast cells, but the transcription of the three enzymes remained similar. Therefore, somehow IMB-D10 inhibited the activity of three chitin synthases and downregulated their protein levels as well. Because chitin synthases were synthesized in the cytoplasm and then transported to the cell wall through vesicles, the binding of this compound to these enzymes might affect their transport to the destination, or this binding might directly destabilize these enzymes, but further work is needed to test these possibilities.

Echinocandins are considered first-line therapy for invasive *Candida* infections [31]. However, treatment failures associated with resistant isolates harboring *fks1* hot-spot mutations have been reported [32]. In recent years, the synergistic effects of nikkomycin and echinocandin have been studied, which suggests that the combination of two antifungal agents can be used in the treatment of infections caused by echinocandin-resistant strains [21]. In this study, we found that IMB-D10 and IMB-F4 had obvious synergistic effects with caspofungin against *S. cerevisiae* and *C. albicans* in vitro. The synergistic effect of these two compounds with echinocandin in vivo will be further studied.

## 4. Materials and Methods

### 4.1. Strains and Culture Conditions

The *S. cerevisiae* strains used in this study were WT BY4741 (*MATa, his3Δ1, leu2, met15∆, ura3-52*), *chs3∆* (*chs3::KanMX* in BY4741), and *fks1∆* (*fks1::KanMX* in BY4741). All the strains were grown in YPD (1% yeast extract, 2% peptone, and 2% glucose) at 30 °C. *C. albicans* (ATCC10231) strains were grown in YPD at 35 °C.

### 4.2. Compound Library Screening

The compound library is a combination of synthetic compounds from Enamine (Kyiv, Ukraine) and Life Chemicals Inc. (Burlington, ON, Canada) and natural products from the Institute of Medicinal Biotechnology. We used WT, *chs3∆,* and *fks1∆* yeast strains to screen new antifungal agents. The screening assays were performed in 96-well plates in a final volume of 200 μL. Fresh yeast cells (OD_600_ = 0.8) were diluted to 10^5^ CFU/mL in YPD medium. We added 198 μL of diluted culture and 2 μL of compounds into each well, and the final concentration of compounds was 100 μg/mL with 1% DMSO. The yeast cells were incubated at 30 °C for 24 hours to assess the growth. For the compounds that inhibited the growth of all three yeast strains, they were serially diluted to determine the minimal inhibit concentration (MIC) with the same method. MIC endpoints were defined as the lowest concentration of drugs that cause a 90% decrease in growth compared to the drug-free control (MIC_90_).

### 4.3. Light and Fluorescence Microscopy

WT *S. cerevisiae* and *C. albicans* ATCC10231 were grown to 10^5^ CFU/mL in YPD medium and then treated with IMB-D10 (12.5 or 25 μg/mL), IMB-F4 (12.5 or 25 μg/mL), or 0.25% DMSO for 3 and 5 hours at 30 ℃. Cells were collected by centrifugation at 1500 g for 5 min and resuspended in PBS. Some cells were examined using 100 × / 1.25 oil immersion objective (DM2500, Leica) and analyzed by Leica las. Some cells were stained with 100 μg/mL Calcofluor White M2R (CFW, Sigma-Aldrich, St. Louis, MO, USA for 3 min and washed three times with water. Then, the cells were examined under the UV light for fluorescent signals.

### 4.4. Analysis of Chitin Synthases Activity

The inhibitory effect of compounds IMB-D10 and IMB-F4, on the activity of chitin synthases 1, 2, and 3, was measured by the procedure, described previously [18]. In brief, WT yeast cells were grown in YPD medium to OD_600_ 1–2. The cells were sedimented at 1500 g for 15 min at 4 °C and washed once with water. Then, the cells were resuspended in cold 100 mM Tris-HCl (pH 7.5) buffer containing protease inhibitors, and broken by 10 cycles of vortexing with acid-washed glass beads (1 min each). The broken cells were centrifuged for 15 min at 1500g to eliminate cell wall and debris. One volume of supernatant was placed on 2 volumes of 10% *w/w* sucrose in 100 mM Tris-HCl pH 7.5 and centrifuged at 55,000 g for 2 hours. The pellet was resuspended in 50 mM Tris-HCl pH 7 and used as membrane protein of Chs3. For Chs1 and Chs2, the membrane protein was pretreated with 80 μg/mL trypsin for 30 min at 30 °C.

To assess the enzyme activity of Chs1, the assay mixture contained 15 μg membrane protein, 5 mM MgCl_2_, 32 mM Tris-maleate (pH6.5), 0.2% digitonin, 40 mM N-acetyl-glucosamine (GlcNAc), and 4 mM UDP-GlcNAc. For Chs2, the mixture contained 15 μg pretreated membrane protein, 1.6 mM CoCl_2_, 32 mM Tris-HCl (pH 8), 40 mM GlcNAc, and 4 mM UDP-GlcNAc. For Chs3, the mixture contained pretreated membrane protein 15 μg, 1.6 mM CoCl_2_, 5 mM NiCl_2_, 32 mM Tris-HCl (pH 8), 40 mM GlcNAc, and 4 mM UDP-GlcNAc. Different concentrations of IMB-D10 and IMB-F4 were added to the mixture, and the reaction without UDP-GlcNAc was used to measure background absorbance. The 96-well plate coated with 50 μg/mL wheat germ agglutinin (WGA) was washed three times with water, and the reaction mixture was added into the wells and incubated at 30 °C for 60 min. The reaction was stopped by 50 mM EDTA, and the plate was emptied and washed with a large volume of water, followed by the addition of 200 μL 0.5 μg/mL WGA-HRP and incubation for 5 min at room temperature. The plate was washed 5 times with running distilled water. Then, the Tetramethylbenzidine (TMB) reagent was added, and the optical density at 600 nm (OD_600_) was immediately detected with Multilabel Reader (Envision, PE, Pontyclun, UK). All reactions were performed in triplicate.

### 4.5. Chitin Content Assay

The chitin content was assessed by measuring the level of D-glucosamine according to the method described by Morgan–Elson [33]. WT yeast cells were grown to 10^5^ CFU/ml in YPD medium and then treated with IMB-D10 (12.5 μg/mL), IMB-F4 (25 μg/mL), or DMSO (0.25%) for 16 hours at 30 °C. Cells were collected by centrifugation at 1500 g for 5 min and then freeze-dried after being washed twice with deionized water. Five-milligram sample was weighted and hydrolyzed by H_2_SO_4_ at 100 °C for 4 hours. The hydrolysates were centrifuged at 12,000 rpm for 5 min. We removed 100 μL supernatant to another tube and adjusted pH value to 3 with 2 M KOH; then the volume was set to 1 mL with water. One milliliter acetylacetone (7% acetylacetone in 1 M Na_2_CO_3_) was added into the tube and incubated for 25 min at 100 °C. The 1.5 mL Ehrlich^’^s reagent and 1.5 mL absolute ethyl alcohol were added into the tube for further incubation for 1 hour at 60 °C. The OD_520_ was determined with BioMet 3S UV-Visible Spectrophotometer (Thermo Scientific^TM^, Waltham, MA, USA). N-acetyl-D-glucosamine was used to establish a standard curve for the quantification of chitin levels. All reactions were performed in triplicate.

### 4.6. Quantitative Real-Time PCR (qRT-PCR)

WT yeast cells were grown to 10^5^ CFU/mL in YPD medium and then treated with 25 μg/mL compounds or 0.25% DMSO for 16 hours at 30 °C. Then, the cells were collected by centrifugation at 1500 g for 5 min and washed twice with water. Total RNA was prepared using the Yeast RNA kit (OMEGA R6870-01, Norcross, GA, USA) and treated with DNase Ⅰ (TransGen GD201-01, Beijing, China) to remove genomic DNA. The resulting RNA was reverse-transcribed with the cDNA synthesis super-mix kit (TransGen AT311, Beijing, China). qRT-PCR was performed using Fast Universal SYBR Green Master (Rox, Roche, Mannheim, Gemany) and gene-specific primers (Appendix A). The DNA level was measured using an Real-Time Quantitative Thermal Cycler (FTC-3000, Funglyn Biotech Inc, Toronto, Canada). The expression levels were correlated with the copy numbers used to amplify the gene to reach the threshold, then represented as the mean ± SEM relative to the cells treated with DMSO. One-way ANOVAs were performed to identify deviation from the mean.

### 4.7. Use Isobaric Tags for Relative and Absolute Quantification (iTRAQ) to Determine the Protein Levels of Chitin Synthases

WT cells were grown to 10^5^ CFU/mL in YPD medium and then treated with 25 μg/mL compounds or 0.25% DMSO for 16 hours at 30 °C. The cells were collected by centrifugation at 1500 *g* for 5 min and then stored at −80 °C. The frozen samples were grounded to a fine powder with liquid nitrogen and then transferred to an Eppendorf tube with phenol extraction buffer containing 1 mM Phenylmethanesulfonyl fluoride (PMSF). The samples were lysed with sonication, and proteins were extracted with phenol and then precipitated with ammonium acetate/methanol. The methanol was excluded by acetone, and the precipitates were collected by centrifugation at 12,000 g for 10 min at 4 °C. The precipitates were dried at room temperature for 3 min and dissolved in SDS lysis buffer for 2 hours. The samples were further centrifuged at 12,000 g for 10 min, and the collected supernatants were centrifuged again to remove precipitation completely. After the determination of protein concentration using Pierce^TM^ BCA Protein Assay kit (Thermo Scientific, Rockford, IL, USA), the protein samples were aliquoted for storage at −80 °C.

The Filter Aided Sample Preparation (FASP) digestion was performed according to a universal sample preparation method for proteome analysis [34]. Peptides were labeled by iTRAQ using iTRAQ Reagents Multiplex Kit (AB SCIEX, Framingham, MA, USA) and then were fractionated by SCX chromatography using the AKTA Purifier system (GE Healthcare, Stockholm, Sweden) [35,36,37]. LC-MS/MS analysis was performed on Nano LC-1D plus (Eksigent, San Francisco, CA, USA) that was coupled to Triple TOF 6600 spectrometer (AB SCIEX, Framingham, MA, USA) for 50 min. The mass spectrometer was operated in positive ion mode. The MS/MS data were analyzed for protein identification and quantification using ProteinPilot software (v.5). The local false discovery rate was estimated with the integrated PSPEP tool in the ProteinPilot Software to be 1% after searching against a decoy concatenated UniProt *S. cerevisiae* protein database. For a protein showing differential expression, it must have been identified and quantified with at least one significant peptide, and the *p*-values of the proteins quantitation should be less than 0.05 and fold change ≥1.5.

### 4.8. Analysis of Synergistic and Antagonistic Effect

Fresh cell cultures of WT *S. cerevisiae* and *C. albicans* ATCC10231 (OD_600_ = 0.8) were diluted to 10^4^ CFU/mL in YPD medium. The in vitro synergistic and antagonistic effects of compounds were determined using checkerboard assay [38]. MIC endpoints were defined as the lowest concentration of drugs, causing 80% decrease in viability compared to the drug-free control (MIC80). The calculation and interpretation of the fractional inhibitory concentration index (FICI) were carried out by referring to the document standards [39]. FICI = (MIC of drug A in combination) / (MIC of drug A alone) + (MIC of drug B in combination) / (MIC of drug B alone). FIC ≤0.5 was considered as a synergistic effect, and >2 was considered as antagonism.

### 4.9. Cytotoxicity Assay

Human embryonic kidney 293 cells and Hela cells were seeded in triplicate in 96-well plates and grown to log-phase in DMEM medium with 10% FBS. Then, the medium was removed, and the cells were incubated with compounds at various concentrations ranging from 3.125 to 100 μg/mL in DMEM medium without FBS. After incubation for 48 hours, MTT reagent was added for further incubation for 4 hours. After we added 50 μL of DMSO, the absorbance was measured at 570 nm. The IC_50_ values were calculated based on a concentration-response curve.

### 4.10. Growth Curve Assay

Fresh cell cultures of WT yeast *S. cerevisiae* and *Candida albicans* ATCC10231 (OD_600_ = 0.8) were diluted in YPD medium (OD_600_ = 0.05). Then, IMB-D10 and IMB-F4 were added at concentrations ranging from 12.5 to 100 μg/mL. The cells were collected at different time points, and OD_600_ was measured with Multi-label Reader. Three experiments were performed independently, and the mean ± SD was calculated.

### 4.11. The Growth Inhibition of S. cerevisiae and C. albicans by Compounds IMB-D10 and IMB-F4

The experiments were carried out using a broth microdilution protocol modified from the Clinical and Laboratory Standards Institute M-27A methods. WT *S. cerevisiae* and *C. albicans* were grown to mid-log phase in YPD medium. The cells were collected by centrifugation at 1500 g for 5 min and diluted with RPMI 1640 medium to the density of 1 × 10^4^ CFU/mL. Then, the samples with various concentrations of compounds were dispensed at 0.2 mL/well in sterile 96-well microplates. The final concentrations of compounds ranged from 3.125 to 100 μg/mL with serial two-fold dilution. After incubation at 30 °C (*S. cerevisiae*) or 35 °C (*C. albicans*) for 24 hours, the minimal inhibitory concentration (MIC) was determined by measuring the optical density of the wells with cell cultures. The antifungal agent caspofungin was used as a positive control. All tests were performed in triplicate. 

### 4.12. Statistic Analysis

All data analyses were performed using GraphPad Prism, version 5, software for Windows (GraphPad Software, San Diego, CA, USA). The *t*-test or one-way ANOVAs was used to determine differences between treatment groups and control. A *p*-value of <0.05 (two-tailed) was considered statistically significant. The data of qRT-PCR were expressed as mean ± SEM values, and the others were mean ± SD values.

## Figures and Tables

**Figure 1 molecules-24-03155-f001:**
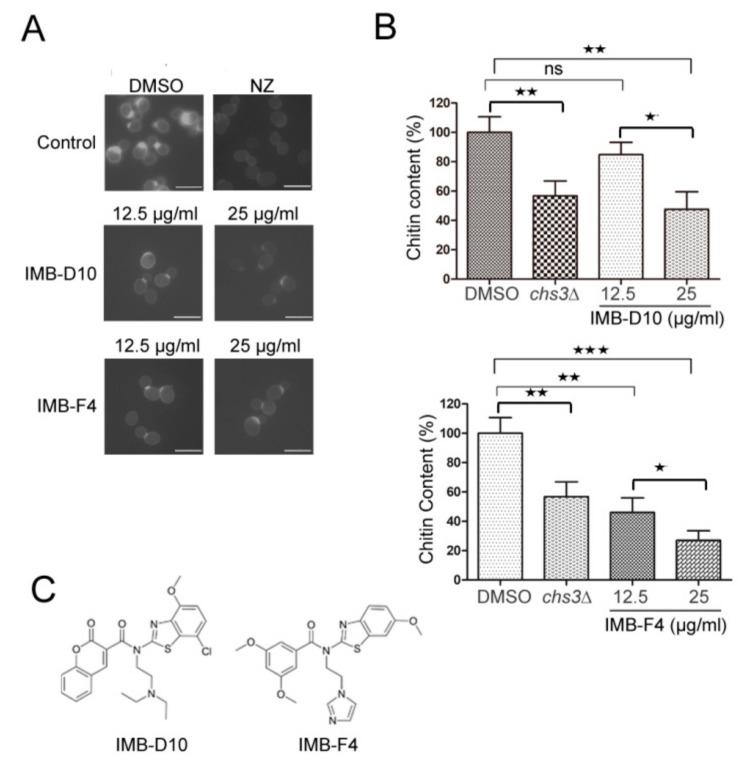
The chitin content in yeast cells treated with compounds IMB-D10 and IMB-F4. (**A**) WT (wild type) yeast cells were treated with IMB-D10 or IMB-F4 for 5 hours and stained with CFW (Calcofluor White M2R). An equal volume of DMSO (0.25%) or 100 μg/mL nikkomycin Z (NZ) was added as a control. The scale bar is 5 μM. (**B**) WT yeast cells were treated with IMB-D10, IMB-F4, or 0.25% DMSO for 16 hours at 30 °C. Yeast mutant cells *chs3∆* cultured for 16 hours were used as a negative control. The chitin content was assessed by measuring the level of D-glucosamine according to the method described by Morgan–Elson. The relative chitin content (%) in cells treated with compounds compared to that in cells treated with 0.25% DMSO was shown. All reactions were carried out in triplicate (mean ± SD; n = 3; * *p* < 0.05, ** *p* < 0.01, *** *p* < 0.001 and ns: no significant difference). *p*-values were calculated with student’s *t*-test. (**C**) The structures of compounds IMB-D10 and IMB-F4.

**Figure 2 molecules-24-03155-f002:**
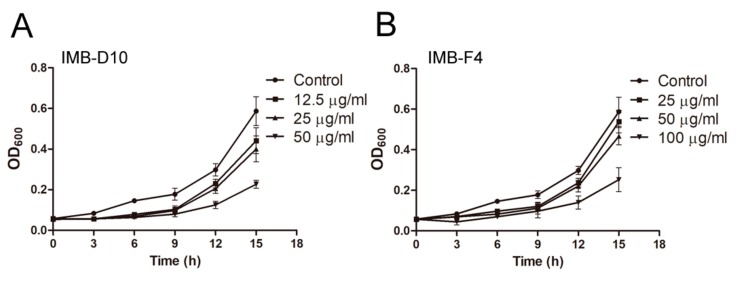
Dose-dependent growth inhibition of WT yeast cells (BY4741) by compounds IMB-D10 (**A**) and IMB-F4 (**B**). Compounds were added into cell cultures to the final concentrations as indicated, and OD_600_ was measured over time. All experiments were performed in triplicate. All cell cultures contained the same concentration of DMSO (0.25%). Data are presented as mean ± SD.

**Figure 3 molecules-24-03155-f003:**
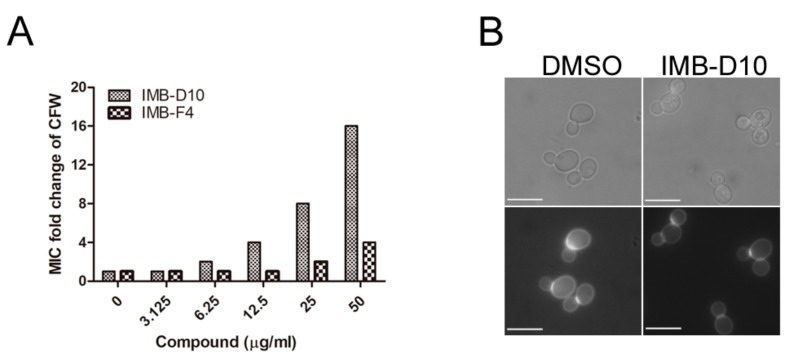
The effect of IMB-D10 and IMB-F4 on the antifungal activity of CFW and cell division. Fresh WT yeast cells were diluted to 10^4^ CFU/ml in YPD (1% yeast extract, 2% peptone, and 2% glucose) medium. (**A**) The MICs (minimum inhibitory concentrations) of CFW to yeast cells were determined in the presence of IMB-D10 or IMB-F4 using checkerboard assay. The fold change of MICs of CFW to yeast cells is shown. All reactions were performed in triplicate, and the same result was obtained. (**B**) WT yeast cells were treated with IMB-D10 (12.5 μg/mL) and IMB-F4 (25 μg/mL) for 5 hours and stained with CFW. All cell cultures contained the same concentration of DMSO (0.25%). The morphology of yeast cells with cell division defect after treatment with IMB-D10 is shown. The scale bar is 5 μm.

**Figure 4 molecules-24-03155-f004:**
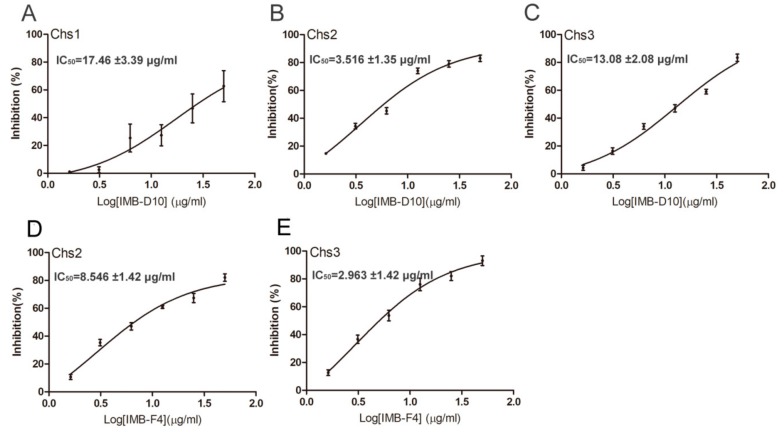
The inhibition of chitin synthesis activity of Chs1, Chs2, and Chs3 by compounds IMB-D10 and IMB-F4. (**A**–**C**). The inhibition of chitin synthase activity by IMB-D10. WT yeast cells were grown in YPD medium to OD_600_ = 1–2. Then, membrane proteins were prepared accordingly and used to analyze the activity of chitin synthases. A 96-well plate coated with 50 μg/mL wheat germ agglutinin (WGA) was used to detect chitin content. The reaction without uridine 5′-diphosphate- N –acetylglucosamine (UDP-GlcNAc) was used to measure the background absorbance. All reactions were carried out in triplicate. Data are presented as mean ± SD, and IC_50_ is shown. (**D** and **E**). The inhibition of Chs2 and Chs3 activity by IMB-F4. The same method was used for the experiments.

**Figure 5 molecules-24-03155-f005:**
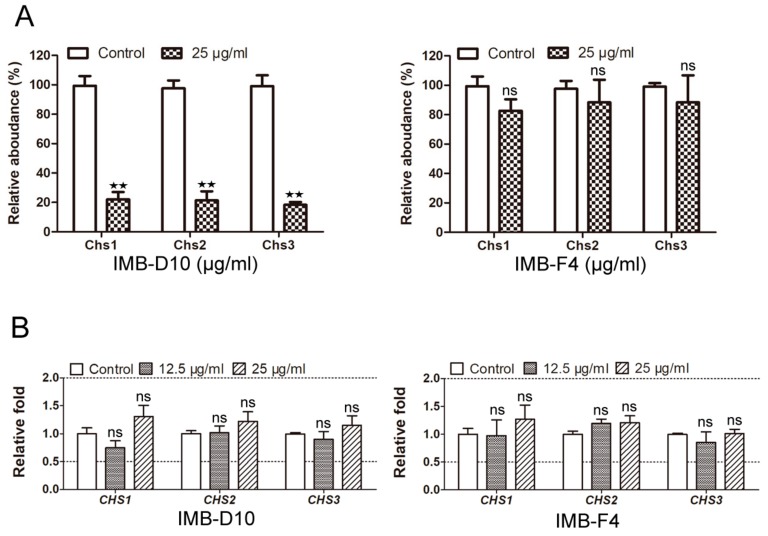
IMB-D10 reduces the protein level of chitin synthases. WT yeast cells were treated with 25 μg/mL IMB-D10 or IMB-F4 for 16 hours. The cells treated with 0.25% DMSO were used as a control. The genome-wide analysis of the relative protein levels was described in the Material and Methods section. All experiments were performed in triplicate. (**A**) The protein level of the three chitin synthases in yeast cells using isobaric tags for relative and absolute quantification (iTRAQ) technology. *p*-values were calculated with student’s *t*-test (mean ± SD; n = 3 ***p* < 0.01 and ns: no significant difference vs. Control). (**B**) The expression levels of *CHS1*, *CHS2,* and *CHS3* genes were analyzed using the qRT-PCR method. *p*-values were calculated with one-way ANOVAs (mean ± SEM; n = 3; ns: no significant difference vs. Control).

**Figure 6 molecules-24-03155-f006:**
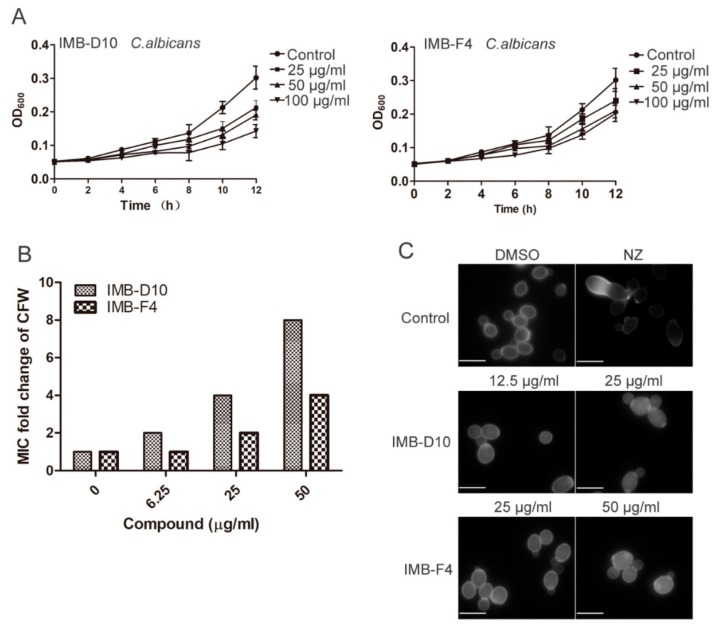
Dose-dependent inhibition of cell growth and chitin synthesis in *C. albicans* by compounds IMB-D10 and IMB-F4. (**A**) IMB-D10 and IMB-F4 inhibit the growth of *C. albicans* in a dose-dependent manner. Data are presented as mean ± SD. (**B**) The MICs of CFW to *C. albicans* in the presence of IMB-D10 or IMB-F4 were determined using checkerboard assay. The fold change of MICs of CFW is shown. (**C**) WT cells were treated with IMB-D10 (12.5 μg/mL) or IMB-F4 (25 μg/mL) for 5 hours and then stained with CFW. An equal volume of DMSO (0.25%) or 100 μg/mL nikkomycin Z (NZ) were used as control. The scale bar is 5μm.

**Table 1 molecules-24-03155-t001:** The MICs (μg/mL) of compounds for WT, *fks1*∆, and *chs3*∆ cells.

	WT	*chs3∆*	*fks1∆*
Nikkomycin IMB-D10	>200 50	>200 100	25 25
IMB-F4	>100	>100	50

MIC is the minimum inhibitory concentration. FICI is a fractional inhibitory concentration index. When FICI ≤0.5, the two drugs were considered as having a synergistic effect. When FICI >2, it was considered as antagonism.

**Table 2 molecules-24-03155-t002:** The synergetic effect of compounds IMB-D10 and IMB-F4 with caspofungin against *S. cerevisiae.*

Compound	MIC (μg/mL)		MIC of Caspofungin (μg/mL)	FICI
	Alone	Combination	Combination	
Nikkomycin	>100	12.5	0.0075	<0.5
IMB-D10	50	12.5	0.0075	0.5
IMB-F4	>100	25	0.0075	<0.5
Caspofungin	0.03	-	-	-

MIC is the minimum inhibitory concentration.

**Table 3 molecules-24-03155-t003:** The synergetic effect of compounds IMB-D10 and IMB-F4 with caspofungin against *Candida albicans.*

Compound	MIC (μg/mL)		MIC of Caspofungin(μg/mL)	FICI
	Alone	Combination	Combination	
Nikkomycin	12.5	3.125	0.00625	<0.5
IMB-D10	100	12.5	0.00625	0.5
IMB-F4	>100	12.5	0.003125	<0.5
Caspofungin	0.025	-	-	-

MIC is the minimum inhibitory concentration. FICI is a fractional inhibitory concentration index. When FICI ≤0.5, the two drugs were considered as having a synergistic effect. When FICI >2, it was considered as antagonism.

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
