# Peer review of "Identification of New Antifungal Agents Targeting Chitin Synthesis by a Chemical-Genetic Method"

_molecules, 2019, doi:10.3390/molecules24173155_

Round 1

Reviewer 1 Report

This manuscript shows interesting data. The concenpt of the study is clear and fine. Therefore, it is acceptable for the jpurnal.

Author Response

Point 1:This manuscript shows interesting data. The concenpt of the study is clear and fine. Therefore, it is acceptable for the journal.

Response 1:Thanks for the positive comments!

Reviewer 2 Report

Li and co-workers present the work entitled “Identification of new anti-fungal agents targeting chitin synthesis by a chemical-genetic method”, which deals with the isolation of compounds that inhibit the activity of chitin synthase in vitro and reduce chitin level in yeast cells, also demonstrating synergistic antifungal effects with caspofungin, and inhibiting the growth of Candida albicans.

The present work shows an interesting approach to the screening for antifungal drugs. However, some Major issues must be addressed in order to validate the results obtained by the authors. Other minor issues should also be addressed in order to improve the clarity, graphical presentation, and overall quality of the manuscript:

MINOR ISSUES:

1. Please pay attention to the fungi genus and species; they should be written in italics. The same for the expression “in vitro”

2. Line 68: “Some antifungals”, instead of “antifungal antibiotics”

3. Line 96: “yeast cells lacking both genes are lethal”. Do the authors mean that the lack of the two genes leads to death? Please rephrase and review the whole paragraph.

4. Table 1: Please correct the table title. As figures, tables should stand by their own, without text support. As so, “The MIC”, is not informative. The meaning of the abbreviation must be displayed, as well as those present in the table, as a footnote. This principle should be applied to all tables of the manuscript.

5. Line 135: Morgan-Elson, and not “Elson-Morgan

6. Figures: All the figures of the manuscript must be revised. Despite being of low resolution, figures legends were pasted in the manuscript embedded in the figure, and not as text, as it should be done.

MAJOR ISSUES:

1. Lack of statistical analysis

This is, indeed, a huge drawback of this manuscript. Taking Figure 1 as an example: The chitin level in yeast cells treated with these two compounds reduced obviously compared to control cells (Figure 1B)”. Authors must make their statements according to a statistical analysis. Without the error bars of the control and the statistical analysis of the results, the observations that seems “obvious” may turn into “significant”, or “not significant”, and thus, without clinical or scientific relevance. The authors have to revise all the results in this way, in order to consolidate their assumptions. As well, and still considering Figure 1, the graphs representing the chitin content does not present standard deviation of the control bars. This information has to be added, once the SD of the control may change all the conclusions by affecting the statistical analysis. The figure legend must give information about the meaning of the error bars (SD or SEM), and the statistical results must be added to the figure, mentioning the values of “p”.

This has to be done for all the results presented in the manuscript.

2. Chemistry of bioactive compounds

Taking into account that the authors are submitting their work to the journal “Molecules”, it would be expected the bioactive compounds to be the central focus of the paper, or at least, have the same importance that was given to the biological activity and mechanistic. Nothing is said about the compounds. The unique mention comes on line 246, where the author say “These two compounds have limited similarities in structure”. Where do they come from? What about their structures? Do they have any similarity with the chitin or glucan inhibitors? What about any structure/activity relationship? From my point of view, this information is mandatory. 

I suggest the authors to start the presentation of the results with the origin, isolation process and chemistry of the compounds IMB-D10 and IMB-F4.

Author Response

Point 1:MINOR ISSUES:

1. Please pay attention to the fungi genus and species; they should be written in italics. The same for the expression “in vitro”

2. Line 68: “Some antifungals”, instead of “antifungal antibiotics”

3. Line 96: “yeast cells lacking both genes are lethal”. Do the authors mean that the lack of the two genes leads to death? Please rephrase and review the whole paragraph.

4. Table 1: Please correct the table title. As figures, tables should stand by their own, without text support. As so, “The MIC”, is not informative. The meaning of the abbreviation must be displayed, as well as those present in the table, as a footnote. This principle should be applied to all tables of the manuscript.

5. Line 135: Morgan-Elson, and not “Elson-Morgan

6. Figures: All the figures of the manuscript must be revised. Despite being of low resolution, figures legends were pasted in the manuscript embedded in the figure, and not as text, as it should be done.

Response 1:Thank you for the suggestions! All these minor issues were addressed in the revised version as suggested. The meaning of the abbreviation has been displayed as a footnote.The figure legends have been moved to the text section.

Point 2:Lack of statistical analysis

    This is, indeed, a huge drawback of this manuscript. Taking Figure 1 as an example: The chitin level in yeast cells treated with these two compounds reduced obviously compared to control cells (Figure 1B)”. Authors must make their statements according to a statistical analysis. Without the error bars of the control and the statistical analysis of the results, the observations that seems “obvious” may turn into “significant”, or “not significant”, and thus, without clinical or scientific relevance. The authors have to revise all the results in this way, in order to consolidate their assumptions. As well, and still considering Figure 1, the graphs representing the chitin content does not present standard deviation of the control bars. This information has to be added, once the SD of the control may change all the conclusions by affecting the statistical analysis. The figure legend must give information about the meaning of the error bars (SD or SEM), and the statistical results must be added to the figure, mentioning the values of “p”.This has to be done for all the results presented in the manuscript.

Response 2:Thank you for your critical comments. As you suggested, the statistical analysis is added in the revised version. The statistical methods are included in the Materials and Methods section as well as in figure legends. The error bars (SD or SEM) and the values of “p” of statistical results are marked in graphs and figures. The results of statistical analysis are described in the revised version. In addition, we repeated the inhibition activity of IMB-D10 and IMB-F4 against S. cerevisiae and Candida albicans. The new growth curves are shown in Fig. 2 and Fig. 6. 

Point 3:Chemistry of bioactive compounds

  Taking into account that the authors are submitting their work to the journal “Molecules”, it would be expected the bioactive compounds to be the central focus of the paper, or at least, have the same importance that was given to the biological activity and mechanistic. Nothing is said about the compounds. The unique mention comes on line 246, where the author say “These two compounds have limited similarities in structure”. Where do they come from? What about their structures? Do they have any similarity with the chitin or glucan inhibitors? What about any structure/activity relationship? From my point of view, this information is mandatory. 

   I suggest the authors to start the presentation of the results with the origin, isolation process and chemistry of the compounds IMB-D10 and IMB-F4.

Response 3:Thank you for your insightful comments. The structure of IMB-D10 is N-(2-(diethylamino) ethyl) -N-(7-hydroxy -4-methoxybenzo [d]thiazol -2-yl) -2-oxo-2H-chromene-3-carboxamide and the structure of IMB-F4 is N-(2-(1H-imidazol-1-yl)ethyl) -3,5- dimethoxy-N-(6-methoxybenzo[d]thiazol -2-yl) benzamide. Both the compounds were from the compound library synthesized by Life Chemicals Inc(Burlington, Canada).At present, there is no report of any activity of these two compounds. These two compounds have limited similarities in structure with benzothiazole, but their activity in the inhibition of chitin synthesis is likely different. IMB-D10, but not IMB-F4, inhibits Chs1 activity and reduces the protein level of the three chitin synthases. In addition, IMB-D10 shows higher inhibitory activity against C. albicans than IMB-F4. These results indicate that their structural differences may have a greater impact on their activity. In our knowledge, compared with IMB-F4, a chlorine substituent on the benzene ring of benzothiazole and a basic aliphatic tertiary amine in the structure of IMB-F10 are likely involved in the differences in activity and mechanism of the two compounds. The in-depth structure-activity relationship of this class of compounds will need more systematically study on the antifungal activity and mechanism of a series of similar compounds. In the new version of this manuscript, we have modified the result and discussion section accordingly.

Reviewer 3 Report

The manuscript of Yan Li et al. described new antifungal agents that inhibit chitin synthesis. The study is aimed to search for new antimicrobial drugs using a new method for the screening. My major comments are:

To evaluate a new method it is necessary that the authors use more molecules and that the statistics must be made on a sufficient number of experiments. Authors should indicate whether the data obtained are significant. The statistic is totally missing. The authors write to have tested the toxicity on Human embryonic kidney 293 cells and Hela cells to the concentration range from 3.125 to 50 μg/ml. Authors should test the molecules at concentrations at least 10 times higher than the active concentration on Candida.

Author Response

Point 1:To evaluate a new method it is necessary that the authors use more molecules and that the statistics must be made on a sufficient number of experiments. Authors should indicate whether the data obtained are significant. The statistic is totally missing.

Response 1:We agree! As you suggested, the statistical analysis is added in the revised version. The statistical methods are included in the Materials and Methods section as well as in figure legends. The error bars (SD or SEM) and the values of “p” of statistical results are marked in graphs and figures. The results of statistical analysis are described in the revised version. In addition, we repeated the inhibition activity of IMB-D10 and IMB-F4 against S. cerevisiae and Candida albicans. The new growth curves are shown in Fig. 2 and Fig. 6.

point 2:The authors write to have tested the toxicity on Human embryonic kidney 293 cells and Hela cells to the concentration range from 3.125 to 50 μg/ml. Authors should test the molecules at concentrations at least 10 times higher than the active concentration on Candida.

Response 2:As suggested, we determined the toxicity of these two compounds to human embryonic kidney 293 cells and Hela cells again. No growth inhibition was noticed when the concentration of these compounds was as high as 100μg/ml. We modified the result in the revised manuscript. These two compounds have poor water solubility and are insoluble at concentrations higher than 100μg/ml. Therefore, we cannot detect their toxicity at higher concentrations.

Round 2

Reviewer 2 Report

The authors have made substantial improvements to the manuscript. However, some minor issues still need to be addressed:

There are still several grammar inaccuracies throughout the manuscript. As so, I suggest a full English revision.

Abstract:

Lines 24-27: From this screen, two compounds were isolated that are toxic to yeast mutants lacking glucan synthase Fks1 but less harmful to wild-type cells and yeast mutants lacking chitin synthase Chs3. The sentence is not grammatically correct. Please rephrase.

The information regarding the origin of the compounds under study, as well as their chemical class, must be added to the abstract.

Introduction: The first paragraph of the introduction lacks references; please review.

Lines 115-116: Again, do the authors mean that the lack of both genes leads to yeast dead? This is different from saying that yeasts are lethal… please rephrase. I sugest the authors to review the use of the word “lethal” throughout all the manuscript.

Lines 173-176: The sentence is not complete…

Line 176: “Both compounds”, instead of “Both the compounds”

4.9 (Cytotoxicity assay): Did the authors incubate the cells with the test compounds, for 48h, in free FBS culture medium? Please clarify.

Author Response

Point 1:There are still several grammar inaccuracies throughout the manuscript. As so, I suggest a full English revision.

Response 1:We went through the manuscript carefully and we noticed many grammar issues. They are fixed in the revised version.

Point 2:

Lines 24-27:From this screen, two compounds were isolated that are toxic to yeast mutants lacking glucan synthase Fks1 but less harmful to wild-type cells and yeast mutants lacking chitin synthase Chs3. The sentence is not grammatically correct. Please rephrase.

The information regarding the origin of the compounds under study, as well as their chemical class, must be added to the abstract.

Introduction: The first paragraph of the introduction lacks references; please review.

Lines 115-116: Again, do the authors mean that the lack of both genes leads to yeast dead? This is different from saying that yeasts are lethal… please rephrase. I sugest the authors to review the use of the word “lethal” throughout all the manuscript.

Lines 173-176: The sentence is not complete…

Line 176: “Both compounds”, instead of “Both the compounds”

Response 2: Thank you for the suggestions! All these minor issues were addressed in the revised version as suggested.

Point 3:(Cytotoxicity assay): Did the authors incubate the cells with the test compounds, for 48h, in free FBS culture medium? Please clarify.

Response 3:The Human embryonic kidney 293 cells and Hela cells were seeded in triplicate in 96-well plates and grown to log-phase in DMEM medium with 10% FBS. Then, the medium were removed and the cells were incubated with compounds in the DEME medium without FBS. After incubation for 48 hours, the growth of cells were detected with MTT assay. We have elaborated this point in more detail in 4.9.

Reviewer 3 Report

The authors improved the manuscript. But there are still some questions.

It is not clear how the antifungal susceptibility test was performed. What is the concentration of DMSO? Why do the authors not use RPMI medium as described in the standard test? 

In the first table how are the MIC values reported (arithmetic mean, median..)? It is not indicated in the caption.

There are still writing errors. Foe example, after the first time Candida must be abbreviated.

Author Response

Point 1:It is not clear how the antifungal susceptibility test was performed. What is the concentration of DMSO?

Response 1: Thank you for your comments. The concentration of DMSO is indicated in the section of Materials and Methods.

Point 2:Why do the authors not use RPMI medium as described in the standard test?

Response 2: In the test of antifungal activity, we used RMPI1640 medium, and our description in the previous version was inaccurate. The description was modified in the section of Material and Method in the revised version. Compared with YPD medium, S. cerevisiae and C. albicans grew slowly in RMPI 1640 medium. Therefore, we first grew these cells in YPD medium. After removing YPD medium by centrifugation, these cells were incubated in 1640 medium. We also determined the MICs of these compounds using YPD or RPMI 1640 media, and the results are the same.

Point 3:In the first table how are the MIC values reported (arithmetic mean, median)? It is not indicated in the caption.

Response 3: When measuring MICs, the compounds were twofold diluted, and the compound concentration of the well where the growth of yeast cells was completely inhibited was considered as the MIC. We repeated the experiments for three times, and the results are the same. Therefore, we did not use mean or other statistical analysis to describe MIC. Other labs also use the same method to determine MICs (Chemical Constituents from the Rhizomes of Smilax glabra and Their Antimicrobial Activity, Molecules 2013, 18, 5265-5287).

Point 4:There are still writing errors. For example, after the first time Candida must be abbreviated.

Response 4: Thank you for your suggestion. We went through manuscript carefully and this issue is fixed in the revised version.